# How memory architecture affects learning in a simple POMDP: the two-hypothesis testing problem

## Abstract

Reinforcement learning is generally difficult for partially observable Markov decision processes (POMDPs), which occurs when the agent's observation is partial or noisy. To seek good performance in POMDPs, one strategy is to endow the agent with a finite memory, whose update is governed by the policy. However, policy optimization is non-convex in that case and can lead to poor training performance for random initialization. The performance can be empirically improved by constraining the memory architecture, then sacrificing optimality to facilitate training. Here we study this trade-off in a two-hypothesis testing problem, akin to the two-arm bandit problem. We compare two extreme cases: (i) the random access memory where any transitions between $M$ memory states are allowed and (ii) a fixed memory where the agent can access its last $m$ actions and rewards. For (i), the probability $q$ to play the worst arm is known to be exponentially small in $M$ for the optimal policy. Our main result is to show that similar performance can be reached for (ii) as well, despite the simplicity of the memory architecture: using a conjecture on Gray-ordered binary necklaces, we find policies for which $q$ is exponentially small in $2^m$, i.e. $q \sim \alpha^{2^m}$ with $\alpha < 1$. In addition, we observe empirically that training from random initialization leads to very poor results for (i), and significantly better results for (ii) thanks to the constraints on the memory architecture.

## 1 Introduction

Reinforcement learning is aimed at finding the sequence of actions that should take an agent to maximise a long-term reward (Sutton & Barto (2018)). This sequential decision-making is usually modeled as a Markov decision process (MDP): at each time step, the agent chooses an action based on a policy (a function that relates the agent's state to its action), with the aim of maximizing its value (the expected discounted sum of rewards). Deterministic optimal policies can be found through dynamic programming (Bellman (1966)) when MDPs are discrete (both states and actions belong to discrete sets) and the agent fully knows its environment (Watkins & Dayan (1992)).

A practical difficulty arises when the agent only have a partial observation of its environment or when this observation is imperfect or stochastic. The mathematical framework is then known as a partially observable Markov decision process (POMDP) (Smallwood & Sondik (1973)). In this framework, the agent's state is replaced by the agent's belief, which is the probability distribution over all possible states. At each time step, the agent's belief can be updated through Bayesian inference to account for observations. In the belief space, the problem becomes fully observable again and the POMDP can thus be solved as a "belief MDP". However, the dimension of the belief space is much larger than the state space and solving the belief MDP can be challenging in practical problems. Some approaches seek to resolve this difficulty by approximating of the belief and the value functions (Hauskrecht (2000); Roy et al. (2005); Silver & Veness (2010); Somani et al. (2013)), or use deep model-free reinforcement learning where the neural network is complemented with a memory (Oh et al. (2016); Khan et al. (2017)) or a recurrency (Hausknecht & Stone (2015); Li et al. (2015)) to better approximate history-based policies.

Here we focus on the idea of Littman (1993), who proposed to give the agent a limited number of bits of memory, an idea that has been developed independently in the robotics community where it is known as a finite-state controller (Meuleau et al. (1999; 2013)). These works show that adding a memory usually increases the performance in POMDPs. But to this day, attempts to find optimal memory allocation have been essentially empirical (Peshkin et al. (2001); Zhang et al. (2016); Toro Icarte et al. (2020)). One central difficulty is that the value is a non-convex function of policy for POMDPs (Jaakkola et al. (1995)): learning will thus generally get stuck in poor local maxima for random policy initialization. This problem is even more acute when memory is large or when all transitions between memory states are allowed. To improve learning, restricting the policy space to specific memory architectures where most transitions are forbidden is key (Peshkin et al. (2001); Zhang et al. (2016); Toro Icarte et al. (2020)). However, there is no general principles to optimize the memory architectures or the policy initialization. In fact, this question is not understood satisfyingly even in the simplest tasks- arguably a necessary step to later achieve a broad understanding.

Here, we work out how the memory architecture affects optimal solutions in perhaps the simplest POMDP, and find that these solutions are intriguingly complex. Specifically, we consider the two-hypothesis testing problem. At each time step, the agent chooses to pull one of two arms that yield random rewards with different means. We compare two memory structures: (i) a random access memory (RAM) in which all possible transitions between $M$ distinct memory states are allowed; (ii) a Memento memory in which the agent can access its last $m$ actions and rewards.

When the agent is provided with a RAM memory, we study the performance of a "column of confidence" policy (CCP): the agent keeps repeating the same action and updates its confidence in it by moving up and down the memory sites until it reaches the bottom of the column and the alternative action is tried. The performance of this policy is assessed through the calculation of the expected frequency $q$ to play the worst arm (thus the smaller $q$, the better). For the CCP, $q$ can be shown to be exponentially small in $M$. This result is closely related to the work of Hellman & Cover (1970) on hypothesis testing and its extension to finite horizon (Wilson (2014)). In practice, we find that learning a policy with a RAM memory and random initialization leads to poor results, far from the performance of the column of confidence policy. Restricting memory transitions to chain-like transitions leads to much better results, although still sub-optimal.

Our main findings concerns the Memento memory architecture. Surprisingly, despite the lack of flexibility of the memory structure, excellent policies exist. Specifically, using a conjecture on Gray-ordered binary necklaces (Degni & Drisko (2007)), we find a policy for which $q$ is exponentially small in $2^m$ —which is considerably better than $q \sim \ln(m)/m$, optimal for an agent that only plays $m$ times. For Memento memory, we also observe empirically that learning is faster and perform better than in the RAM case.

The code to reproduce the experiments is available at `https://anonymous.4open. science/r/two-hypothesis-BAB3`, and uses a function defined here `https:// anonymous.4open.science/r/gradientflow/gradientflow`. The experiments where executed on CPUs for about 10 thousand CPU hours.

## 2 POMDPs and the two-hypothesis testing problem

### 2.1 General formulation

**Definition 2.1** (POMDP). A discrete-time POMDP model is defined as the 8-tuple $(S, A, T, R, \Omega, O, p_0, \gamma)$: $S$ is a set of states, $A$ is a set of actions, $T$ is a conditional transition probability function $T(s'|s, a)$ where $s', s \in S$ and $a \in A$, $R : S \to \mathbb{R}$ is the reward function[1], $\Omega$ is a set of observations, $O(o|s)$ is a conditional observation probability with $o \in \Omega$ and $s \in S$, $p_0(s) : S \to \mathbb{R}$ is the probability to start in a given state $s$, and $\gamma \in [0, 1)$ is the discount factor.

A state $s \in S$ specifies everything about the world at a given time (the agent, its memory and all the rest). The agent starts its journey in a state $s \in S$ with probability $p_0(s)$. Based on an observation $o \in \Omega$ obtained with probability $O(o|s)$ the agent takes an action $a \in A$. This action causes a transition to the state $s'$ with probability $T(s'|s, a)$ and the agent gains the reward $R(s)$. And so on.

---

[1]In the literature, $R$ also depends on the action: $R : S \times A \to \mathbb{R}$. Our notation is not a loss of generality. The set of state can be made bigger $S \to S \times A$ in order to contain the last action.

**Definition 2.2** (Policy). A policy $\pi(a|o)$ is a conditional probability of executing an action $a \in A$ given an observation $o \in \Omega$.

**Definition 2.3** (Policy State Transition). Given a policy $\pi$, the state transition $T_\pi$ is given by

$$T_\pi(s'|s) = \sum_{o,a \in \Omega \times A} T(s'|s,a)\pi(a|o)O(o|s). \tag{1}$$

**Definition 2.4** (Expected sum of discounted rewards). The expected sum of future discounted rewards of a policy $\pi$ is

$$G_\pi = \mathbb{E}_{\substack{s_0 \sim p_0 \\ s_1 \sim T_\pi(\cdot|s_0) \\ s_2 \sim T_\pi(\cdot|s_1) \\ \cdots}} \left[ \sum_{t=0}^{\infty} \gamma^t R(s_t) \right]. \tag{2}$$

Note that a POMDP with an expected sum of future discounted rewards with discount factor $\gamma$ can be reduced to an undiscounted POMDP (Altman (1999)), as we now recall (see proof in Appendix A):

**Lemma 2.1.** *The discounted POMDP defined in 2.1 with a discount $\gamma$ is equivalent to an undiscounted POMDP with a probability $r = 1 - \gamma$ to be reset from any state toward an initial state. In the undiscounted POMDP, the agent reaches a steady state $p(s)$ which can be used to calculate the expected sum of discounted rewards $G_\pi = \frac{1}{r}\mathbb{E}_{s \sim p}[R(s)]$.*

## 2.2 OPTIMIZATION ALGORITHM

To optimize a policy algorithmically, we apply gradient descent on the expected sum of discounted rewards. First, we parametrize a policy with parameters $w \in \mathbb{R}^{|A| \times |\Omega|}$, normalized to get a probability using the softmax function $\pi_w(a|o) = \frac{\exp(w_{ao})}{\sum_b \exp(w_{bo})}$. Then, we compute the transition matrix $\tilde{T}_\pi$, from which we obtain the steady state $p$ using the power method (See Appendix B). Finally, we calculate $G_\pi$ by the Lemma 2.1.

Using an algorithm that keeps track of the operations (we use `pytorch` Paszke et al. (2017)), we can compute the gradient of $G_\pi$ with respect to the parameters $w$ and perform gradient descent with adaptive time steps (i.e. a gradient flow dynamics):

$$\frac{d}{dt}w = \frac{d}{dw}G_\pi(w). \tag{3}$$

## 2.3 TWO-HYPOTHESIS TESTING PROBLEM

The problem we consider is the two-hypothesis testing problem. We label two arms by the letters A and B. The two arms gives a reward of $+1$ or $-1$ with a Bernoulli distribution. The probabilities to obtain a positive reward are noted $k_A$ and $k_B$ respectively. The environment is entirely defined by the couple $(k_A, k_B)$. With equal probability, the environment is in one of the following two configurations (hypothesis):

$$\begin{cases} k_A = \frac{1+\mu}{2} & k_B = \frac{1-\mu}{2} & \text{(hypothesis } H_A) \\ k_A = \frac{1-\mu}{2} & k_B = \frac{1+\mu}{2} & \text{(hypothesis } H_B) \end{cases} \tag{4}$$

where the hypothesis $H_A$ (resp. $H_B$) corresponds to A (resp. B) being the best arm. In expectation over the environments, an agent that plays randomly or always the same arm will have a reward 0.

Note that this problem is similar to the Bandit problem, except that in the latter $(k_A, k_B)$ can take any value in the square $[0,1]^2$. When $r \to 0$, our results below can be generalized to the bandit problem, as done in Cover & Hellman (1970) by recasting the latter as finding the correct hypothesis ( 'Arm A is better' or 'Arm B is better').

In our setup, the agent only knows its last arm played, reward obtained (if there were some) and the state of its memory (different memories are described below). Based on that, it chooses an arm (play A or B) and how to update its memory state.

In the POMDP formalism (c.f. 2.1), the state $s$ contains the environment $(k_A, k_B)$, the memory state, the last arm played and reward obtained. We only consider agents that have a complete access

to their memory, therefore $O$ is deterministic and simply projects $s$ by removing the environment information.

$s$ = environment $H_A$ or $H_B$), memory state, last arm played (A or B) and last reward (1 or $-1$)

$o$ = memory state, last arm played and last reward

$a$ = arm to play, memory update

$$(5)$$

We define the function $q(s)$, which is 1 if the agent just played the "wrong" arm in state $s$, and 0 if he played the correct one. The probability to play the wrong arm is then $q_\pi = \mathbb{E}_{s \sim p}[q(s)]$ where $p$ is the steady state of the problem with reset $r$. The expected sum of discounted gains $G_\pi$ can be related to $q_\pi$ as: $G_\pi = \frac{\mu}{r}(1 - 2q_\pi)$. In the following, we will use $q_\pi$ as a measure of performance, trying to find a policy $\pi$ that minimizes $q_\pi$ (and thus maximizes $G_\pi$).

## 2.4 TYPES OF MEMORY CONSIDERED

**Definition 2.5** (Random Access Memory (RAM)). RAM is the most flexible memory setting. The agent has $M$ memory states and has full control over it. It has $|A| = M \times 2$ possible actions: the choice of the next memory state and which arm to play. There is a high degeneracy in the space of strategies since any permutation of the memory states leads to the same performance. Note that, since our agent can use the information of the last arm and reward, the total number of memory states is in fact $M_{\text{eff}} = 4M$, which corresponds to $2 + \log_2 M$ bits.

**Definition 2.6** (Memento Memory[2]). The agent only has access to the information of its past $m$ actions and rewards. For instance, for $m = 4$, an observation could be `AABB++-+`. We use the notation of the most recent action/reward on the right (here, the last action was B and the reward was $+1$). If the agent plays A and obtains a positive reward, the next memory state would be `ABBA+-++`. In this memory architecture, the agent writes in its memory only through the plays he does ($|A| = 2$). Here, the number of bits is $2m$ and the total number of memory states is in fact $M_{\text{eff}} = 4^m$.

## 3 GAIN AND EXPLORATION WITH A RANDOM ACCESS MEMORY

Hellman & Cover (1970); Cover & Hellman (1970) described an optimal policy for the RAM architecture in the limit of a small reset $r$. In their optimal policy, the memory states $i = 1...M$ are organized linearly with transitions only occurring between $i$ and $i - 1$ (if the last observation supports $H_A$) or $i + 1$ (if the last observation supports $H_B$). In the limit $r \to 0$, transition probabilities can be shown to be independent on $i$ for $1 < i < M$. The two extreme memory states $i = 1$ and $i = M$ are special as they present a vanishing exit rate $\epsilon \to 0$. Thus, only these two states are visited with finite probability in that limit. For such a policy, one obtains an optimal probability to play the worst arm $q_{\text{HC}}(M) = \alpha^{M-1}/(\alpha^{M-1} + 1)$, with $\alpha = (1 - \mu)/(1 + \mu)$. [3]

Our set-up is slightly different, as we allow for the choice of arm to depend on both the memory state and the information of the last arm played and reward obtained (Figure 1**A-B**). In that case, the policy can be improved, as demonstrated by considering the *column of confidence* policy.

**Definition 3.1** (column of confidence policy (CCP)). It is a RAM policy with $M$ memory states. It is depicted in Figure 1**C**. Essentially, the agent uses its last arm played to effectively increase the size of its memory by a factor 2.

The probability $q$ to play the worst arm by following CCP is derived in Appendix C for general $r$, by writing the transition probability matrix $T_\epsilon$ with a generic $\epsilon$, for any of the two hypotheses $H_A$ or

---

[2]*Memento* is a Christopher Nolan's film where the hero has a short-term memory loss every few minutes. Using photos and tattoos, the hero keeps track of information he will eventually forget, thus encoding information into his own actions.

[3]To see why a linear policy is optimal, introduce $\lambda_i = P(H_A|i)/P(H_B|i)$, with $P(H_A|i)$ the conditional probability that $H_A$ is true if the memory state $i$ is visited. Choose the labels $i$ such that $\lambda_1 \geq \lambda_2... \geq \lambda_M$. Then it can be shown that $\lambda_i/\lambda_{i+1} \leq \alpha^{-1}$ (Hellman & Cover (1970)). The linear policy saturates this bound for all $i$, leading to the maximal ratio that can be obtained between any two states $\lambda_1/\lambda_M = \alpha^{1-M}$. This ratio controls the gain in the limit $\epsilon \to 0$ where only these two states are visited with finite probabilities.

Figure 1: **A** Update scheme from Cover & Hellman (1970) vs **B** our update scheme. Using the same notation as Cover & Hellman (1970), $y_n$ is the reward obtained by playing the arm $e_n$ and $T_n$ is the memory state (not to confuse with the transition probability $T$ of Sec 2.1). To make the link with Sec 2.1 in our setup the state $s_n$ correspond to the tuple $(e_n, y_n, T_n, k_A, k_B)$, the observation $o_n$ to the triplet $(e_n, y_n, T_n)$ and the action $a_n$ is represented by the purple and green arrows. The red arrow can be seen as part of the conditional probability $T(s_{n+1}|s_n, a_n)$. **C** The column of confidence policy, that can be used in the RAM case, exemplified for $M = 8$. The memory states are organized into two columns. The distribution $p_0$ initializes the agent's memory into states $1A$ or $1B$ with equal chance. The agent keeps playing the same arm, moving up and down a column depending on the reward, unless it is in the memory state 1 (it then switches arm after a negative reward). Once the agent reaches a state at the top of a column, it can only step down with small probability $\epsilon$ if a negative reward is obtained. This two-column arrangment effectively double the number of memory states, by creating $2M = 16$ distinct states. In this policy, the value of the memory can be viewed as a measure of the confidence in the arm being played.

$H_B$. From the stationary distribution $T_\epsilon \vec{p} = \vec{p}$, one obtains $q(\epsilon)$, which reaches its minimum value for:

$$\epsilon = \sqrt{2} \frac{\sqrt{\alpha - \alpha^{3M-1} + \alpha^M(2\mu - 3) + \alpha^{2M}(2\mu + 3)}}{\sqrt{1 - \alpha^M(1 - \alpha^M - \mu - \alpha^M \mu)}} \sqrt{r} + \mathcal{O}(r) \qquad (6)$$

The result $\epsilon \sim \sqrt{r}$ indicates a non-trivial balance between exploration and exploitation, in which the time spent in each extreme state of the memory grows only as the square root of the horizon time $1/r$. A similar result was obtained for hypothesis testing (Wilson (2014)).

For $r \to 0$ (a result generalized for any $k_B$ and $k_A$ in Appendix C), we obtain:

$$q_{\text{CCP}}(M) = \frac{\alpha^{2M-1}}{\alpha^{2M-1} + 1}. \qquad (7)$$

Note that it is the optimal gain of a RAM memory of size $2M$. Since our agent memory size is $M_{\text{eff}} = 4M$ (the factor 4 coming from the 2 possible past actions and two possible rewards), the results of Hellman & Cover (1970); Cover & Hellman (1970) show that CCP is nearly optimal, in the sense that no policies with one less bit of memory can do better. In Figure 2, we confirm empirically that it is at least a local policy optimum, as performing policy optimization near this solution leads to no further improvements.

## 4 GAIN FOR MEMENTO MEMORY

Are there efficient policies when the agent memorizes the last $m$ arms played and rewards obtained (Memento memory cf. 2.6)? In the classical two-arm bandit problem, after a time $m$, the optimal strategy selects the worst arm with probability $q = O(\ln m/m)$ (Auer et al. (2002)). It turns out that

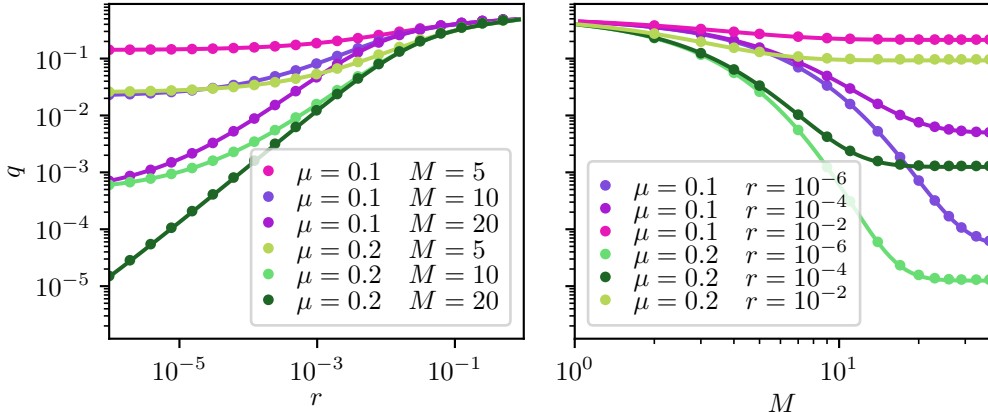

Figure 2: **Left** Probability to play the worst arm $q$ *vs.* the reset probability $r$ for different memory sizes $M$ and two values of $\mu$ (the difference between the mean outcome of the two arms). **Right** $q$ *vs.* memory size $M$ for different $r$ and $\mu$ indicated in legend. The solid lines show the analytical result corresponding to column of confidence policy (CCP), whereas symbols show the results of the learning algorithm (see Sec 2.2) for a RAM memory with a policy initialized close to the predicted optimal column of confidence policy ($\pi_0 \approx \pi_{\mathrm{CCP}} + 10^{-4}$). Learning does not find strategies that perform better than the optimal column of confidence policy, even for large values of $r$, indicating that it is a local optimum. In this figure, error bars would be smaller than the symbols.

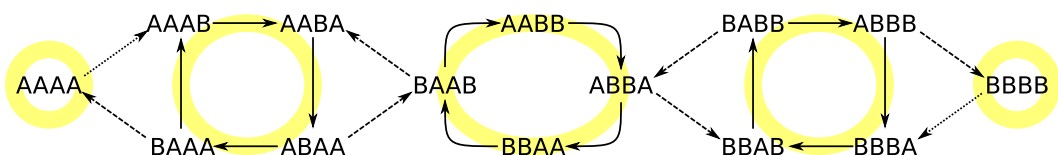

Figure 3: Necklace policy with memory of the $m = 4$ last arms played and rewards obtained (Memento memory cf. 2.6). Memory states are organized into 3 (5 if we consider the two end states) cycles of arms played, called necklaces in combinatorics. The memory state also contains the rewards obtained, but these are not explicitly shown here for the sake of readability. Most of time, the agent stays in the same necklace by playing the oldest action he remembers. Each necklace has 2 inputs and 2 outputs states (some states can be both input and output). The agent has a finite probability to leave its current necklace only when the output state is maximally informative: all 4 rewards are $+$ for A and $-$ for B to move to the left, or the opposite to move to the right. Thicker arrows represent high probability transition (deterministic in some situations); dashed arrows represent input/output transitions between necklaces; and thin arrows represent the small probabilities to leave the two end states.

our agent can use his own actions to encode events over a time much longer than $m$, leading to $q$ exponentially small in $2^m$ in a stationary state with long horizon $r \to 0$.

**Definition 4.1** (Necklace policy). The necklace policy is based on 4 key ingredients (Figure 3).

(i) Most of the time, the agent plays the oldest action in its memory (i.e. the arm played $m$ actions before). Doing so, it memorizes actions cycle inside binary necklaces of length $m$ (in combinatorics, a necklace is an equivalent class of character strings under cyclic rotation, here the strings are words of length $m$ made of the letters A and B, hence binary). When $m$ is prime, there are exactly $N(m) = 2 + (2^m - 2)/m$ distinct necklaces. For any $m$, the number of necklaces can be derived from Pólya (1937)'s enumeration theorem and is equal to $N(m) = \frac{1}{m} \sum_{d|m} \varphi(d) 2^{m/d}$, where $\varphi$ is the Euler's totient function.

(ii) We provide a Gray order on the necklaces. It means that necklaces are numbered and two successive necklaces can only differ by one letter. We order the necklaces from $i = 1$ for the necklace where all actions are A to $i = n(m)$ for the necklace where all actions are B. The necklace $i = 1$ (resp. $i = n(m)$) also corresponds to the maximum confidence in hypothesis $H_A$ (resp. $H_B$). In general, the longest possible chain of necklace, $n(m)$, is unknown and less than the total number $N(m)$ of necklaces. But, when $m$ is prime, it has been conjectured (and checked for $m \leq 37$) that there exists a Gray order of all distinct necklaces (Degni & Drisko (2007)): in other words, $n(m) = N(m) = 2 + (2^m - 2)/m$, for $m$ prime.

(iii) The probability to exit a necklace is zero, except for two exit configurations for which this probability is $\epsilon_1 > 0$ if two conditions are met. First, the memorized actions must allow the agent to switch from the necklace $i$ to the necklace $i - 1$ or $i + 1$ by taking a new action. Second, the sequence of rewards must be maximally informative: to switch to the necklace $i - 1$ (i.e. gaining confidence in $H_A$), all rewards have to be $+1$ for the arm A and $-1$ for the arm B and the opposite to switch to the necklace $i + 1$.

(iv) In the two extreme states, the probability of exit is $\epsilon_0$ when all rewards are negative. Below, we consider the limit $\lim_{\epsilon_1 \to 0} \lim_{\epsilon_0 \to 0} q(\epsilon_0, \epsilon_1)$ of this strategy. This order of limits ensures that only the extreme states are visited with a finite probability and that the agent cycles many times in each necklace before exit.

In order to compute the optimal gain of the necklace policy we introduce the following two lemmas.

**Lemma 4.1.** *Assume a discrete random walk on a chain of sites indexed by $i$, with probabilities $r_i$ to step from $i$ to $i + 1$ and $l_i$ to step from $i$ to $i - 1$. Starting in site $i$, the probability to reach site $j + 1$ ($i \leq j$) before site $i - 1$ is*

$$P_{i \to j} = \left( 1 + \frac{l_i}{r_i} + \frac{l_i}{r_i} \frac{l_{i+1}}{r_{i+1}} + \cdots + \frac{l_i}{r_i} \cdots \frac{l_j}{r_j} \right)^{-1}. \tag{8}$$

*Proof.* The proof is developed in Appendix D.1. □

**Lemma 4.2.** *Assume a discrete random walk on a chain of $n + 2$ sites indexed by $i = 0, 1, \ldots n + 1$ (with probabilities $r_i$ to step to the right and $l_i$ to the left). If $r_0 = \epsilon R$ and $l_{n+1} = \epsilon L$, in the limit $\epsilon \to 0$, the probability to be on site $n + 1$ is*

$$p(n + 1) = \left( 1 + \frac{L}{R} \prod_{i=1}^{n} \frac{l_i}{r_i} \right)^{-1}. \tag{9}$$

*Proof.* The proof is developed in Appendix D.2. □

Using these two lemma the following theorem can be proved.

**Theorem 4.3.** *Consider the two-hypothesis problem described in* (4) *in the limit of small reset $r \to 0$. The necklace policy described in Definition 4.1 with parameters $(\epsilon_0, \epsilon_1)$ has a probability $q$ to play the worst arm satisfying $q \geq q^*$, with:*

$$q^* = \left( 1 + \alpha^{m(1 - n(m))} \right)^{-1} \quad \text{with } \alpha = \frac{1 - \mu}{1 + \mu}. \tag{10}$$

*The optimal necklace policy converges to $q^*$ when $r \ll \epsilon_0 \ll \epsilon_1 \ll 1$.*

*Proof.* The detailed proof is contained in Appendix D.3. □

**Note** At leading order $q^* = \alpha^{2^m} + o(\alpha^{2^m})$. This is because the number of distinct necklaces $N(m)$ only differs from $2 + (2^m - 2)/m$ at second order, and because we expect to find Gray orders within those distinct necklaces whose length only differ from $N(m)$ at second order.

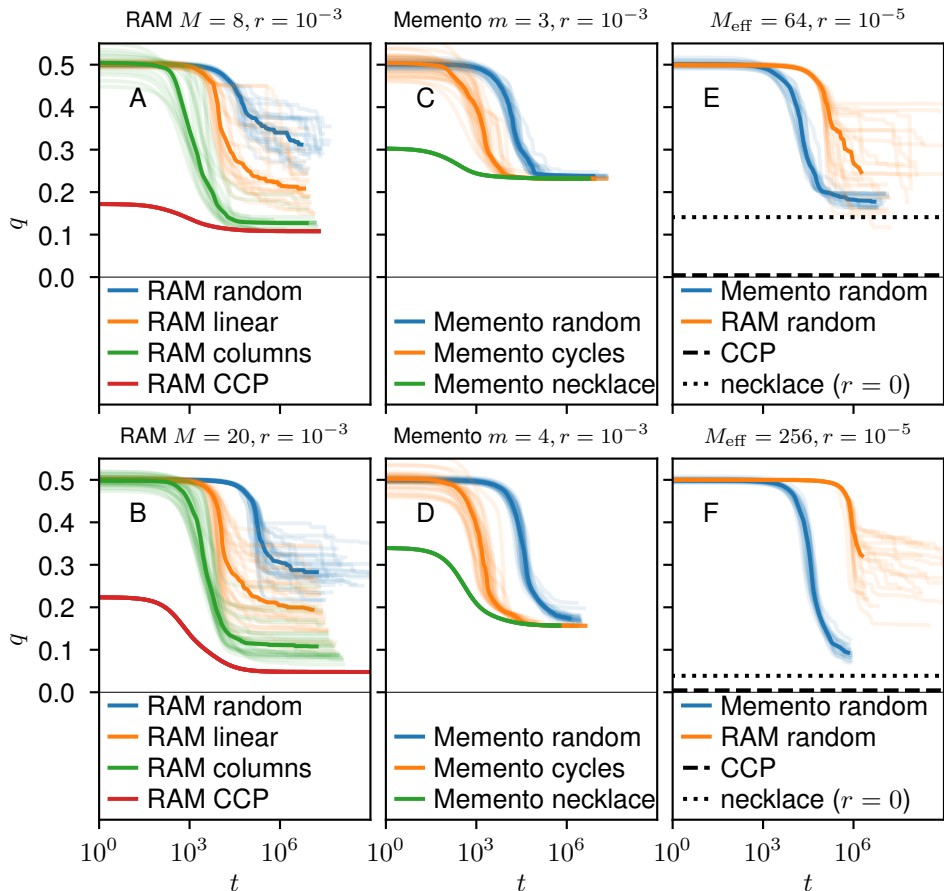

Figure 4: Dynamics of the optimization algorithm for different initialization methods. Probability to play the worse arm $q$ *vs.* algorithm time $t$ (time defined in (3)) for different seeds. 20 initialization seeds are shown in light color and the median is shown in solid color. The wall time is caped to 1 hour per optimization. **A-B** RAM with $M = 8$ and $M = 20$, we compare the *random* initialization, the *linear* initialization where the jumps in memory states are initialized to be contiguous, the *columns* initialization corresponding to linear initialization with the extra constraint that the last action is repeated except for the memory state 1 and the *CCP* initialization, very close to the column of confidence policy (i.e. $\pi_0 \approx \pi_{\text{CCP}} + 10^{-4}$, a difference that explains why the red curves can decrease). **C-D** $q$ *vs.* $t$ for the Memento memory $m = 3$ and $m = 4$. We compare the *random* initialization with the *cycles* initialization that repeats the oldest action, except if all the remembered plays correspond to a maximally informative event (during training a path between the cycles has to be learned) and the *necklace* where $\epsilon_0$ and $\epsilon_1$ has to be learned. **E-F** $q$ *vs.* $t$ for randomly initialized policies. The values of $m$ (3 and 4) and $M$ (16 and 64) are chosen such that the total memory needed to perform these strategies $M_{\text{eff}}$ is identical in each panel. The dashed line corresponds to calculation of $q$ for the CCP and for the necklace policy (for which we only have a prediction for $r = 0$). In this figure $\mu = 0.1$.

## 5 POLICY OPTIMIZATION AND LOCAL MINIMA

To study empirically how learning depends on the memory architecture, we measure how the probability $q(t)$ to play the worse arm after a training time $t$ depends on the initialization of policy. For the RAM memory, we find that random initialization (blue curves) leads to very poor results (panels A and B of Figure 4). Results however improve when a linear structure for memory states is imposed (orange curves) and when the arms played are segregated on the two sides of that linear structure

to form two columns (green). However, even in that case, training does not converge towards the optimal column of confidence parameters, unless parameters are initialized near the optimal values (red curves).

By contrast, training with the Memento memory (consisting in the last $m$ actions and rewards, cf. 2.6) appears less sensitive to initialization. As shown in the panel C and D of Figure 4, initializing the policy randomly (blue) performs does not perform as well as initializing the policy with necklaces (orange), however the difference is not significant.

Although the RAM architecture is in principle more flexible (and in fact include Memento memories), we find that, for random initialization, the Memento architecture leads to actually *better* policies after training. The comparison is shown in panels E and F of Figure 4, where the two memory architectutes are compared keeping the effective memory size $M_{\text{eff}}$ constant. This finding emphasizes the need to constrain memory architecture, so as to obtain smoother optimization landscapes.

## 6 CONCLUSION

| | Policies | |
|---|---|---|
| Memory scheme | Memento (cf. 2.6) | RAM (cf. 2.5) |
| Policy | necklace (cf. 4.1) | CCP (cf. 3.1) |
| Effective memory | $M_{\text{eff}} = 4^m$ | $M_{\text{eff}} = 4M$ |
| Performance ($q^{-1} - 1$) | $\alpha^{m(1-n(m))} \sim \alpha^{-2^m}$ | $\alpha^{-(2M-1)}$ |
| ... as function of $M_{\text{eff}}$ | $\sim \alpha^{-\sqrt{M_{\text{eff}}}}$ | $\sim \alpha^{-M_{\text{eff}}/2}$ |
| ... more generally for $(k_A > k_B)$ | $\left(\frac{1-k_B}{1-k_A}\right)^m \left(\frac{1/k_B-1}{1/k_A-1}\right)^{m(n(m)-2)/2}$ | $\frac{1-k_B}{1-k_A}\left(\frac{1/k_B-1}{1/k_A-1}\right)^{M-1}$ |

Table 1: Comparison of the performances of the necklace and CCP strategies in the limit $r \to 0$. As shown in the last line, the gain of these policies can be generalized to any distribution of the Bernoulli probabilities $k_A$ and $k_B$ (but needs not be optimal then).

Our results are summarized in Table 1 that compares the necklace policy (cf. 4.1) and the column of confidence policy (cf. 3.1). For each of these policies, we provide the optimal performance, reached in the limit $r \to 0$. We conjecture that these policies are the optimal ones for the Memento and RAM memory schemes respectively. Concerning the Memento memory, this conjecture is supported by the simulations shown in Figure 3: the best numerical policies found for $m = 3$ and $m = 4$ are in fact the necklace policy.

An interesting additional questions for the future is the generalization of these ideas to a broader set of tasks. The CCP appears well-suited for multiple hypotheses testing (Chandrasekaran & Lakshmanan (1978); Yakowitz et al. (1974)), where it would correspond to a "star" policy with a branch for each hypothesis. Classifying optimal policies for more complex hierarchical tasks, such as those involved in navigation (Theocharous et al. (2004); Toussaint et al. (2008)), would have practical applications. Looking ahead, it would be interesting to understand if these ideas have applications to other approaches dealing with POMDPs, including recurrent networks (Li et al. (2015)) whose theoretical understanding remains very limited.

Finally, it is intriguing that for all memory structures studied, a linear organization of memory states appears to be optimal. Despite the fact that our set-up is intrinsically digital, optimal policies approach an analog memory architecture with a single degree of freedom: it corresponds to the position along the chain, and measures the relative belief of one hypothesis over the other. In neuroscience, dominants models of decision making often present a single analogue variable being updated by observations (Gold & Shadlen (2007); Rescorla & Wagner (1972)). It would be interesting to test experimentally, in situations where the environment can change with a small probability $r$ between two distinct classes, if animals stick to two extreme believes, and leave them for exploration with some rate $\sim \sqrt{r}$.

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

# Appendices

## A  PROOF OF LEMMA 2.1

*Proof.* The state transition matrix of the undiscounted POMDP is

$$\tilde{T}_\pi(s'|s) = rp_0(s') + (1-r)T_\pi(s'|s). \tag{11}$$

The steady state $p(s)$ has to be stable through $\tilde{T}_\pi$, thus satisfying $p(s') = \sum_s \tilde{T}_\pi(s'|s)p(s)$. In the tensor form, it can be written $\vec{p} = r\vec{p}_0 + (1-r)T_\pi\vec{p}$. Applying recursively this formula $n$ times we obtain:

$$\vec{p} = r\sum_{t=0}^{n-1}(1-r)^t T_\pi^t \vec{p}_0 + (1-r)^n T_\pi^n \vec{p}. \tag{12}$$

Translating it into an expectation value expression we obtain:

$$\mathbb{E}_{s\sim p}[f(s)] = r\,\mathbb{E}_{s_t\sim T_\pi^t p_0}\left[\sum_{t=0}^{n-1}(1-r)^t f(s_t)\right] + (1-r)^n\mathbb{E}_{s\sim T_\pi^n p}[f(s)] \tag{13}$$

for any function $f$ and where $T_\pi p$ is understood as a matrix-vector product (note that $T_\pi p \neq p$). By replacing $f$ by $R$ and by taking the limit $n \to \infty$, for $r = 1 - \gamma$, we can identify (13) with (2), thus obtaining the Lemma 2.1. $\square$

## B  IMPLEMENTATION DETAILS

To compute the steady state we use the power method algorithm: Alg.1.

When we have multiple independent environments (by environment we mean subset of $S$ that the agent cannot escape with its actions), $S$ is the disjoint union of these environments: $S = S_1 + S_2 + \dots$. If $p_0$ factorize as follow $p_0(s) = P(S_i)P(s|S_i)$ for $s \in S_i$, we can compute the steady state by computing those of each independent environments.

The initial state of the memory of the agent can be optimized by allowing gradient flow to modify specific part of $p_0$. It could mathematically be reformulated as special actions done on special initial states to initialize the memory.

**Data:** A transition matrix $M$
**Result:** The steady state
**while** *the columns of $M$ differ (with a given tolerance)* **do**
 | $MM \to M$;
**end**
**return** a column of $M$;

**Algorithm 1:** Power method

## C  EXACT COMPUTATION FOR COLUMN OF CONFIDENCE POLICY

The *mathematica* notebook is provided along with the code, see the link in Section 1.

Here we compute the optimal value of $\epsilon$ (that maximize the gain) given $r$ and $\mu$.

First observe that the states (3B+, 1A-, ...) can be arranged along a line:

$$\begin{array}{ccccccccc} \text{MA+} & \dots & \text{2A+} & \text{1A+} & \text{1B+} & \text{2B+} & \dots & \text{MB+} \\ \text{MA-} & \dots & \text{2A-} & \text{1A-} & \text{1B-} & \text{2B-} & \dots & \text{MB-} \end{array} \tag{14}$$

where the probability transition only occurs between two consecutive states.

If we merge the $\pm$ we have $2M$ states and we can compute their transition matrix without reset:

$$Q = \begin{bmatrix} 1-\epsilon k_B & k_A & & & & \\ \epsilon k_B & 0 & k_A & & & \\ & k_B & 0 & \ddots & & \\ & & k_B & \ddots & k_A & \\ & & & \ddots & 0 & \epsilon k_A \\ & & & & k_B & 1-\epsilon k_A \end{bmatrix} \tag{15}$$

where $k_A = (1+\mu)/2$ and $k_B = (1-\mu)/2$

Including the reset, the transition probability becomes

$$M_{ij} = (1-r)Q_{ij} + rJ_i \tag{16}$$

where $J$ is $1/2$ in the two central states `1A` and `1B`.

The steady state $p_i$ is the solution of

$$\sum_j M_{ij}p_j = p_i \iff \sum_j ((1-r)Q_{ij} - \delta_{ij})p_j = -J_i \tag{17}$$

We can split this problem in 5 regions: the A border, the A bulk, the center, the B bulk, the B border. In the bulks the (17) is

$$(1-r)k_B p_{i-1} - p_i + (1-r)k_A p_{i+1} = 0 \tag{18}$$

The solutions are of the form $p_i = c_1 w_+^i + c_2 w_-^i$ with

$$w_\pm = \frac{1 \pm \sqrt{1 - (1-r)^2(1-\mu^2)}}{(1-r)(1+\mu)} \tag{19}$$

To fix the coefficients $c_1, c_2$ in the two bulks we have 6 equations:

- two on the left border (two first lines of (17))
- two in the center, at the injection (two middle lines of (17))
- two on the right border (two last lines of (17))

Solving these equations, we can compute the probability to play the wrong arm (B, assuming $\mu > 0$).

$$q = \frac{\begin{array}{c} w_+(1+w_+)(1-w_-)w_-^{2M}(1+w_+(\epsilon-1))(w_-+\epsilon-1) \\ +(w_+ \leftrightarrow w_-) \\ +(w_-w_+)^M(w_-w_+-1)((w_--1)(w_+-1)(w_-+w_+)(\epsilon-1)+2w_-w_+\epsilon^2) \end{array}}{2(w_--w_+)(w_+^{2M}w_-(1+w_-(\epsilon-1))(w_++\epsilon-1) - (w_+ \leftrightarrow w_-))} \tag{20}$$

in this expression of $q$ we expressed $r$, $\mu$ and $k_A$, $k_B$ in function of $w_\pm$

Optimizing $q$ with respect to $\epsilon$ leads to

$$\epsilon = (w_-w_+)^{-M} \frac{-\sqrt{\dfrac{(w_--1)w_-^M(w_+-1)w_+^M(w_-w_+-1)(w_-^M w_+ - w_- w_+^M)^2}{(w_--1)(w_+-1)(w_-w_+)^{1+2M}(w_-w_+-1)^2(w_+^M-w_-^M)}\left(\begin{array}{c} w_-^{3M}w_+^2(1+w_-)(1+w_+) \\ -(w_+ \leftrightarrow w_-) \\ +w_+^{2M}w_-^{1+M}(2w_++w_-+w_+w_-(7+2w_+)+w_-^2(w_+-1)) \\ -(w_+ \leftrightarrow w_-) \end{array}\right)}}{(w_-w_+-1)\left(\begin{array}{c} w_-^2 w_+^{2M}(1+w_-(w_+-1)+w_+) \\ +(w_+ \leftrightarrow w_-) \\ -2w_-^{1+M}w_+^{1+M}(1+w_-w_+) \end{array}\right)} \tag{21}$$

We can make the Taylor expansion of $\epsilon$ with respect to $r$

$$\epsilon = \sqrt{2}\frac{\sqrt{\alpha - \alpha^{3M-1} + \alpha^M(2\mu - 3) + \alpha^{2M}(2\mu + 3)}}{\sqrt{1 - \alpha^M(1 - \alpha^M - \mu - \alpha^M\mu)}}\sqrt{r} + \mathcal{O}(r) \tag{22}$$

with $\alpha = \frac{1-\mu}{1+\mu}$

For $M$ large, it converge quickly toward

$$\epsilon = \sqrt{\frac{2r}{1 - \mu^2}} + \mathcal{O}(r) \tag{23}$$

In the limit $r \to 0$ we get

$$q = \frac{\alpha^{2M-1}}{\alpha^{2M-1} + 1} + \mathcal{O}(\sqrt{r}) \tag{24}$$

## D   RANDOM WALK ALONG A CHAIN

### D.1   PROBABILITY TO TRAVERSE THE CHAIN

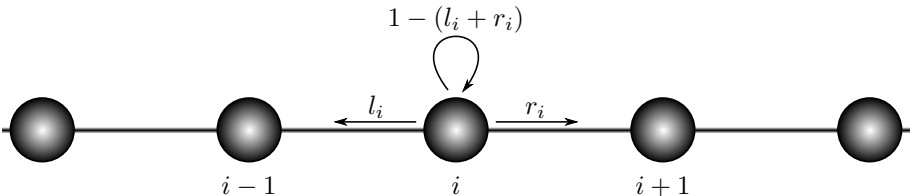

Figure 5: Markov chain

Consider a random walk on a chain of sites with probabilities $r_i$ to move from site $i$ to $i + 1$ and $l_i$ to move from $i$ to $i - 1$ (Figure 5). First, we want to compute the probability $P_{i \to j}$ (with $i \le j$) that, starting at site $i$, the walker visits the site $j + 1$ before the site $i - 1$. Note that $P_{i \to j}$ only depends on $\{l_k\}_{k=i}^j$ and $\{r_k\}_{k=i}^j$. Note also that according to this definition $P_{i \to i} = \frac{r_i}{l_i + r_i}$. We also define $P_{i \leftarrow j}$ as the probability to reach $i - 1$ before $j + 1$ by starting in $j$, then we have $P_{i \leftarrow i} = \frac{l_i}{l_i + r_i}$.

Starting in $i$, after one time step the walker is either (i) in $i - 1$ (with probability $l_i$) and the probability to reach $j + 1$ before reaching $i - 1$ becomes null, (ii) still in $i$ (with probability $1 - (l_i + r_i)$) and the probability to each $j + 1$ before $i - 1$ is still given by $P_{i \to j}$, (iii) in $i + 1$ (with probability $r_i$). Once in $i + 1$ there are two possibilities: either the walker never comes back to $i$ and it reaches $j + 1$ (with probability $P_{i+1 \to j}$), or it does (with probability $1 - P_{i+1 \to j}$) and it again has the same probability $P_{i \to j}$ to reach $j + 1$ before $i - 1$. Thus we have:

$$P_{i \to j} = (1 - (l_i + r_i))P_{i \to j} + r_i\left(P_{i+1 \to j} + (1 - P_{i+1 \to j})P_{i \to j}\right). \tag{25}$$

The quantities that matter are $p_i \equiv r_i/(l_i + r_i)$. If we isolate $P_{i \to j}$ to the l.h.s. we get

$$P_{i \to j} = \frac{p_i P_{i+1 \to j}}{1 - p_i(1 - P_{i+1 \to j})}. \tag{26}$$

Making the changes of variable $X_{i \to j} \equiv \frac{1 - P_{i \to j}}{P_{i \to j}}$ and $x_i \equiv \frac{1 - p_i}{p_i}$ (note that $x_i = l_i/r_i$) we obtain

$$X_{i \to j} = x_i(1 + X_{i+1 \to j}). \tag{27}$$

By repeating the formula we see that we get

$$X_{i \to j} = \sum_{k=i}^j \prod_{l=i}^k x_l = x_i + x_i x_{i+1} + x_i x_{i+1} x_{i+2} + \cdots + x_i \cdots x_j \tag{28}$$

from which we can get $P_{i \to j}$ by the inverse transformation

$$P_{i \to j} = \left(1 + \frac{l_i}{r_i} + \frac{l_i}{r_i}\frac{l_{i+1}}{r_{i+1}} + \cdots + \frac{l_i}{r_i} \cdots \frac{l_j}{r_j}\right)^{-1}. \tag{29}$$

## D.2 CHAIN WITH FINAL STATES

Let us consider the same chain as above with a finite length $n$ such that sites are labelled from $i = 1$ to $n$. Now let us add a final state at each end of the chain (sites $i = 0$ and $n + 1$). We consider the probabilities to leave the final states asymptotically small, of order $\epsilon$. More precisely, the probability to move from $i = 0$ to 1 is $\epsilon R$ (we call it $R$ as it is a probability to go to the Right, although it concerns the most left site) and the probability to move from $n + 1$ to $n$ is $\epsilon L$. These probability and the probabilities $p(0)$ and $p(n + 1)$ to be on the site 0 and $n + 1$ are related by a balance of the flow from 0 to $n + 1$:

$$p(0)\epsilon R P_{1 \to n} = p(n + 1)\epsilon L P_{1 \leftarrow n}. \tag{30}$$

In the limit $\epsilon \to 0$, the probabilities to be in the extreme states tends to 1 and we thus have $p(0) = 1 - p(n + 1)$, yielding

$$p(n + 1) = \left( 1 + \frac{L}{R} \frac{P_{1 \leftarrow n}}{P_{1 \to n}} \right)^{-1}. \tag{31}$$

This result can be simplified using the equality

$$\frac{P_{1 \leftarrow n}}{P_{1 \to n}} = \frac{1 + \frac{l_1}{r_1} + \cdots + \frac{l_1}{r_1} \cdots \frac{l_n}{r_n}}{1 + \frac{r_n}{l_n} + \cdots + \frac{r_n}{l_n} \cdots \frac{r_1}{l_1}} = \prod_{i=1}^{n} \frac{l_i}{r_i}, \tag{32}$$

to finally obtain the formula

$$p(n + 1) = \left( 1 + \frac{L}{R} \prod_{i=1}^{n} \frac{l_i}{r_i} \right)^{-1}. \tag{33}$$

## D.3 CHAIN OF NECKLACES

To compute the gain of the necklace policy, the idea is to show that necklaces can be arranged on a chain so that we can use (33) (see Figure 3).

For $m$ prime, the number of non trivial necklaces is $(2^m - 2)/m$, the trivial necklaces being the two final states (i.e., the words AAA..A and BBB..B). Using the conjecture of Degni & Drisko (2007), there is a Gray order on these necklaces when $m$ is prime. In other words, there is a chain from one final state to the other that passes exactly once by each necklace, the difference between two successive necklaces being exactly one bit (i.e. a single A is changed into B or vice versa).

In any case ($m$ prime or not), we call $n(m)$ the length of the longest chain with Gray order. We call $y(m)$ the length of the smallest necklace in that longest chain (arguably the smallest prime factor of $m$).

A necklace is characterized by the numbers $a$ and $b$ of letters A and B in the m-long word, with $a + b = m$. After at least one complete loop in the necklace (i.e. at least $y(m)$ actions), the probabilities to leave that necklace (when we are at the exit states) are

$$l_i = \epsilon_1 k_A^a (1 - k_B)^b, \tag{34}$$
$$r_i = \epsilon_1 (1 - k_A)^a k_B^b. \tag{35}$$

With the odds to do at least one loop increasing as $\epsilon_1$ goes to zero or $y(m)$ goes to $\infty$.

In the two final states, and again after one loop, the probabilities to leave are $\epsilon_0 L = \epsilon_0 (1 - k_B)^m$ and $\epsilon_0 R = \epsilon_0 (1 - k_A)^m$. We can now use the formula (33) (when $\epsilon_0 \ll \epsilon_1 \ll 1$), in which the

product simplifies as

$$\prod_{i=1}^{n} \frac{l_i}{r_i} = \prod_{i=1}^{n(m)-2} \frac{k_A^{a_i}(1-k_B)^{b_i}}{(1-k_A)^{a_i}k_B^{b_i}} \tag{36}$$

$$= \frac{\prod_{i=1}^{n(m)-2}(1/k_B - 1)^{b_i}}{\prod_{i=1}^{n(m)-2}(1/k_A - 1)^{a_i}} \tag{37}$$

$$= \frac{(1/k_B - 1)^{\sum_i b_i}}{(1/k_A - 1)^{\sum_i a_i}} \tag{38}$$

$$= \left(\frac{1/k_B - 1}{1/k_A - 1}\right)^{m(n(m)-2)/2} \qquad \text{since } \sum_{i=1}^{n(m)-2} (a_i + b_i) = m(n(m) - 2) \tag{39}$$

where $a_i$ and $b_i$ are the occurrences of A and B in the necklace $i$.

Inserting (39) into (33) and using the value of $k_A$ and $k_B$ given in (4) for hypothesis $H_A$ leads to

$$p(n+1) = \left(1 + \alpha^{m(1-n(m))}\right)^{-1}, \qquad \text{with } \alpha = \frac{1-\mu}{1+\mu}. \tag{40}$$

and $p(0)$ can be obtained by changing $\mu$ into $-\mu$ or $\alpha$ into $1/\alpha$, by symmetry of the necklace policy. The probabilities under hypothesis $H_B$ are obtained by exchanging $p(0)$ and $p(n+1)$, again by symmetry.

Under hypothesis $H_A$ (resp. $H_B$), the value $p(n+1)$ (resp. $p(0)$) corresponds to the probability $q^*$ to play the worst arm if the probabilities of the non-final necklaces are zero (which is asymptotically true if $\epsilon_0 \ll \epsilon_1$). To reach $q^*$, we also need $\epsilon_1$ to be asymptotically small in order to guarantee at least one loop in each necklace and $\epsilon_0$ has to be asymptotically larger than the reset $r$. In summary, we need $r \ll \epsilon_0 \ll \epsilon_1 \ll 1$ to reach asymptotically $q^*$, otherwise the probability $q$ will be larger than $q^*$.

### D.4 COLUMN OF CONFIDENCE POLICY WITH NO RESET

When there is no reset $r = 0$ we can compute the performance of the column of confidence policy for two arms of probabilities $k_A$ and $k_B$. We obtain via (33) (assuming $k_A > k_B$)

$$q^{-1} - 1 = \frac{1 - k_B}{1 - k_A} \left(\frac{1/k_B - 1}{1/k_A - 1}\right)^{M-1} \tag{41}$$

