# OpenReview forum: "How memory architecture affects learning in a simple POMDP: the two-hypothesis testing problem"
_ICLR.cc/2022/Conference — ICLR 2022 Submitted_

### Official Review · Reviewer_BH7m · 2021-10-29

**Correctness:** 4
**Technical Novelty And Significance:** 2
**Empirical Novelty And Significance:** 2
**Recommendation:** 3
**Confidence:** 3

**Main Review:**

Strengths:

1. The work is in an interesting direction and points out potential issues with current empirical approaches to learning memory allocations.

2. The explanation of optimal policies (such as the necklace policy) and their proofs were well-done and insightful.

Weaknesses:

1. While the authors cite shortcomings in recent works when it comes to learning memory allocations, it would be nice to see the ideas explored here scaled up to similar-sized problems. If the principle of "restricted memory architectures are easier to learn with and more robust to random initializations" is true, this should be testable in larger and more complex POMDP's (and policies) even without knowing the optimal policies.

A simple/naive way to do this would be to compare something like RNN's to something like an attention-based architecture that can only observe its last m interactions with the environment (or any other restrictive memory architecture). These could be trained with reinforcement learning in a visual navigation environment, for example. The size of m or the recurrent state could then be played with.

Performing experiments like this would not be particularly difficult and would easily make the insights of the paper applicable to modern techniques.


**Summary Of The Paper:**

The authors compare two different memory architectures for the two-hypothesis testing problem. They show that a simple fixed memory where the agent can only access the previous m environment interactions is more robust to random initializations than a more flexible RAM-setup while still maintaining similar performance.

**Summary Of The Review:**

This work points out potential issues with current empirical approaches to learning memory allocations and suggests that restricted memory architectures could be easier to use. While the authors do show this to be the case in one very simple environment, they do not show that this concept scales to other larger or more complex environments, such as visual navigation, that are currently used. As-is, the work is simply too narrow in scope and experiments to be a significant contribution to modern techniques and understanding.

---

### Official Review · Reviewer_PEeX · 2021-11-01

**Correctness:** 4
**Technical Novelty And Significance:** 2
**Empirical Novelty And Significance:** 2
**Recommendation:** 5
**Confidence:** 2

**Main Review:**

The problem studied in the paper is quite a bit of interest for strategy learning and synthesis for POMDPs where, unlike MDPs, memory is known to be essential.

While the methodology in the paper is rigorous and explained well, the scope is very narrow to derive any general insight. The comparison is limited to a single problem.

While the results are stated relatively carefully, there is room for improvement and better precision. For example, what is the meaning of "<<" in Theorem 4.3?

Memory in POMDPs has attracted interest in neighboring contexts and from a different perspective in RL recently. The paper seems to be missing that literature.
https://arxiv.org/pdf/2007.08351.pdf
https://arxiv.org/pdf/2009.11459.pdf
https://arxiv.org/abs/2105.14073

**Summary Of The Paper:**

The paper compares two memory structures for policies for POMDPs through theoretical and empirical analysis in a relatively simple problem.

**Summary Of The Review:**

An interesting yet very limited paper.

---

### Official Review · Reviewer_7ywF · 2021-11-04

**Correctness:** 2
**Technical Novelty And Significance:** 2
**Empirical Novelty And Significance:** 2
**Recommendation:** 3
**Confidence:** 3

**Main Review:**

First, even though the ideas in this paper are fairly intuitive, I admit I found this paper very hard to read. Many of the explanations are not clear and the overall writing can be substantially improved.

The main novelty, in my opinion, is the presentation of the Gray-ordered necklaces, which I find to be very nice and quite original (while the CCP policy is well studied). However, the theoretical analysis of this structure is lacking, in the sense that the authors only analyzed this specific structure, but gave no theoretical guarantees on its optimality in the class of the Memento-based memories. At most, the authors conjecture optimality based on simulations that might only imply a local optimum. From a practical perspective, I think that this idea does not scale. First, it's not clear how to scale it to more than two arms, nor to non-binary rewards. Second, as a concept, playing actions in a cyclic manner can be disastrous in dynamic environments, so the usefulness of this idea beyond bandit problems is unclear.

Moreover, the connection of the analyzed model to general POMDPs is a bit artificial, and I don't really see what insights the paper gives on memory architectures in POMDP. If I understand correctly, the claim is that Memento works better than RAM with random initialization, even though some RAM structures are better than Memento. This claim is mainly based on the empirical evaluation, and for it to be meaningful, it should be methodologically tested on various POMDP of different characteristics (e.g., such that require long/short memory, fast/slow mixing, etc.). Moreover, the application of both memory architectures is somewhat naive, and as both were extensively studied, I think that this does not suffice to prove this claim.

Other comments
- p.g. 2 - "the value is a non-convex function of policy for POMDPs... This problem is even more acute when memory is large or when
all transitions between memory states are allowed" - please refrain from such comments. The value in nonconvex in the policy also in MDPs, but it is still possible to find an optimal solution - it's all about the structure of the problem and the chosen representation, so using it to claim that 'more memory is bad' is not really meaningful. For example - what if some RAM structure effectively reconstructs the real state?
- POMDP formulation - the formulation in this paper feels very tailored for the specific problem and algorithms, but it's not the standard formulation. Specifically, it's not very natural to put the memory state inside the state representation - the memory is a characteristic of the policy, not of the environment, and for some representations, it unnecessarily increases the environment exponentially.
- Honestly, I didn't understand section 2.2 and the related appendix (B). Please improve the explanations and add more details about how you calculate everything.
- In Fig. 4, right column, it seems like the RAM-random option did not converge yet - does it converge to a worse value or just converge more slowly?

**Summary Of The Paper:**

This paper tackles a two-armed bandit problem (of means $Ber(1/2+\mu)$ and $Ber(1/2-\mu)$, respectively) when memory is limited. Specifically, they model this problem as a POMDP (or, alternatively, hypothesis testing), where the hidden state determines the mean of each arm. The author claim that this simple model might provide some insights on memory architecture in general POMDPs (namely, how the history representation affects the learning process). The authors describe two memory architectures:

(i) A general finite-state machine ('RAM'), which transitions based on the played arm and its reward. Specifically, they focus on the column of confidence policy (CCP). The memory structure - a chain where each end of the chain represents an arm. $+1$ reward moves the state towards an arm and $-1$ moves it away, except for the end states that have low escape probability. The policy - playing the arm that its end-state closer to the current state.

(ii) History is presented as a memory buffer with the last arm plays and their rewards('Memento'). In this case, the authors present a policy that plays the same action sequence cyclically ('necklaces'). These cycles are chained in a Gray-ordering and the policy moves between cycles (change one action) only if a full cycle indicates that this switch is beneficial.

The authors analyze the (stationary) probability of playing the worse actions and show that the RAM model enjoys better (lower) probability. Finally, the authors try to learn a policy when relying on either RAM or Memento architectures. They showed that a random initialization, the Memento memory performs better, while imposing structural constraints that correspond with the suggested policies in (i) and (ii) make RAM memory perform better.

**Summary Of The Review:**

The paper presents an interesting algorithm for two-armed bandit problems with finite history, but the ideas do not scale up to larger problems, and the empirical evaluation is not enough to draw any conclusion on more general POMDPs.

---

### Official Review · Reviewer_42t7 · 2021-11-07

**Correctness:** 2
**Technical Novelty And Significance:** 3
**Empirical Novelty And Significance:** 2
**Recommendation:** 3
**Confidence:** 3

**Main Review:**

How memory structure affects POMDP learning is very interesting, and it could be useful for future design of learning algorithm under uncertainties. But the paper only focuses on a very simple problem with two possible hypotheses for the unknown POMDP model. Though  insights gained from studying a simple problem could be useful for general problems, the paper doesn't provide discussion on how the ideas from the simple problem could be applied in general problems. Given the policy and analysis for the simple two-hypothesis problem, is there any insight which could be useful for the general problem? Could an agent with RAM or memento memory in a general problem learn the unknown parameter with some simulations? Any feature of the proposed policies may be extended to general POMDP problems?

For the simple two-armed problem, two memory structures are considered: random access memory and memento memory. For each memory structure, one policy is proposed with performance analysis. The proposed policy itself is not particularly novel since they are  available in the literature. The analysis is limited that the provided performance results are only in the asymptotic region when the discount factor converges to one. Given that the analysis focuses on the limit when the discount factor approaches one, why not considering the average performance formulation? Another import analysis would be the performance of the algorithm in a finite time horizon. Would it be possible to have some finite time performance analysis?

There are also some issues in the presentation and lack of details. Section 3 states that empirical results for local search are shown in Figure 2; however, Figure 2 only shows the results under different mu and r, and no results for the suggested local optimization. For policy optimization with different initialization schemes, there is no details on how each initialization scheme is implemented. Are the memory schemes only different in their initialization or actually different in their update dynamics? Is the same learning algorithm is used with different memory initialization or different policies are used to adapted to the memory structure?

**Summary Of The Paper:**

The paper attempts to investigate how memory architecture affects learning performance of POMDP agents. It focuses on a very simple two-arm bandit problem with two hypotheses for their probabilities. Two memory structures are considered: random access memory and memento memory. For each memory structure, one policy is provided with a kind of asymptotic optimal performance. Simulation results for simple gradient based learning algorithms are shown under the two memory structures.

**Summary Of The Review:**

The considered problem is a bit too simple to have meaningful insights, and the paper provides almost no discussion on how the analysis on this simple problem could be useful for general POMDP problems. Even for the simple problem setting, the paper doesn't have performance guarantees for the proposed policy in the non-asymptotic region.

---

### Decision · Program_Chairs · 2022-01-20

**Decision:**

Reject

**Comment:**

All reviewers agreed that the contribution is too limited for the paper to be published. I encourage the authors to take the reviews into account when improving their work.